# Cost–Utility Analysis of Open Radical Hysterectomy Compared to Minimally Invasive Radical Hysterectomy for Early-Stage Cervical Cancer

**DOI:** 10.3390/cancers15174325

**Published:** 2023-08-29

**Authors:** Nadav Michaan, Moshe Leshno, Gil Fire, Tamar Safra, Michal Rosenberg, Shira Peleg-Hasson, Dan Grisaru, Ido Laskov

**Affiliations:** 1Gynecologic Oncology Department, Tel Aviv Sourasky Medical Center, Sackler School of Medicine, Tel Aviv University, Tel Aviv 6423906, Israel; gilf@tlvmc.gov.il (G.F.); michalrosenberg@gmail.com (M.R.); grisaro@post.tau.ac.il (D.G.); idol@tlvmc.gov.il (I.L.); 2Gastro-Enterology, Tel Aviv Sourasky Medical Center, Sackler School of Medicine, Tel Aviv University, Tel Aviv 6423906, Israel; leshnom@tauex.tau.ac.il; 3Oncology Department, Tel Aviv Sourasky Medical Center, Sackler School of Medicine, Tel Aviv University, Tel Aviv 6423906, Israel; tamars@tlvmc.gov.il (T.S.); shira.hasson@rmh.nhs.uk (S.P.-H.)

**Keywords:** cervical cancer, minimally invasive surgery, open radical hysterectomy, cost–utility, QALY

## Abstract

**Simple Summary:**

Patients with early-stage cervical cancer treated by minimally invasive surgery show shorter progression-free and overall survival compared to open surgery. This study integrated minimally invasive and open surgery survival data with surgery costs and utilities and conducted a cost-effectiveness analysis, using a Markovian decision analysis model, to compare the two surgical approaches. Our results show that open radical hysterectomy is not only oncologically superior but also more cost-effective. Until new data regarding the safety of minimally invasive surgery and surgery costs are presented, open radical hysterectomy should be the preferred approach from both the oncological and financial standpoint.

**Abstract:**

We aimed to investigate the cost-effectiveness of open surgery, compared to minimally invasive radical hysterectomy for early-stage cervical cancer, using updated survival data. Costs and utilities of each surgical approach were compared using a Markovian decision analysis model. Survival data stratified by surgical approach and surgery costs were received from recently published data. Average costs were discounted at 3%. The value of health benefits for each strategy was calculated using quality-adjusted life years (QALYs). Incremental cost-effectiveness ratio, calculated using the formula (average cost minimal invasive surgery—average cost open surgery)/(average QALY minimal invasive surgery—average QALY open surgery), was used for cost-effectiveness analysis. One-way sensitivity analysis was conducted for all variables. Open radical hysterectomy was found to be cost-saving compared to minimally invasive surgery with an incremental cost-effectiveness ratio of USD −66 and USD −373 for laparoscopic and robotic surgery, respectively. The most influential parameters in the model were surgery costs, followed by the disutility involved with open surgery. Until further data are generated regarding the survival of patients with early-stage cervical cancer treated by minimally invasive surgery, at current pricing, open radical hysterectomy is cost-saving compared to minimally invasive radical hysterectomy, both laparoscopic and robotic.

## 1. Introduction

Following the introduction of laparoscopic radical hysterectomy in the 1990s, this surgical approach gained favor and the number of minimally invasive surgeries for early-stage cervical cancer increased substantially around the world [1,2,3]. While the advantages of open surgery over laparoscopic surgery may include shorter surgery length, less need for dedicated equipment, and different surgical expertise required, the advantages of laparoscopic surgery over open surgery are well-documented and include lower intra-operative blood loss as well as lower rates of post-operative complications, such as wound infections, fever, sepsis, and ileus, in addition to lower post-operative pain and shorter hospital stay [1,2,4,5,6,7,8,9]. Despite the lack of quality prospective data to support the use of minimally invasive surgery in the treatment of early-stage cervical cancer, this approach was widely adopted, and in many oncological centers, it has become the standard procedure.

In 2018, the first prospective, randomized trial comparing the oncological safety of minimally invasive surgery and open surgery for the treatment of early-stage cervical cancer was published by Ramirez et al. The Laparoscopic Approach to Cervical Cancer (LACC) trial showed inferior oncological outcomes, including shorter progression-free survival and overall survival for the minimally invasive approach [10]. Along with the LACC trial, several other epidemiological studies, retrospective cohorts, and meta-analyses were published, showing inferior oncological outcomes for women with early-stage cervical cancer treated by minimally invasive surgery, compared to open surgery [11,12,13,14,15,16,17,18,19]. Since the publication of the LACC trial, the number of minimally invasive surgeries has significantly decreased in centers around the world, and if it still used, it is saved mainly for patients with small lesions under 2 cm [20,21,22].

Despite the many advantages of minimally invasive surgery, the total costs of laparoscopic, and particularly robotic radical hysterectomy, have been consistently shown to exceed those of open surgery, mainly due to higher instrument- and procedure-related costs [1,2,23]. Previous studies comparing the cost-effectiveness of open surgery and minimally invasive surgery for the treatment of endometrial cancer show that even though minimal invasive surgery entails higher costs compared to open surgery, laparoscopic hysterectomy is in fact cost-effective, with higher quality-adjusted life years (QALYs) gained for the minimally invasive approach [24,25]. Minimally invasive radical prostatectomy was also found to be more cost-effective compared to open surgery for prostate cancer. This calculation was sensitive to the number of surgeries performed by each center per year [26].

In light of the updated survival data from the prospective randomized LACC trial, the aim of our study was to perform a cost–utility analysis of open radical hysterectomy compared to minimally invasive surgery for the treatment of early-stage cervical cancer, using USA pricing data.

## 2. Materials and Methods

As this work is a theoretical, mathematical/financial model, not involving any human or animal subjects in any form, after consultation with our local institutional review board, it was exempt from the need for IRB approval.

In order to compare the costs and utilities of open radical hysterectomy and minimally invasive radical hysterectomy for early-stage cervical cancer, a Markov model was constructed using TreeAgePro software (TreeAge™ Software LLC. version 2022). Additional analyses were performed using MATLAB (MathWorks^®^ 2020b, Natick, MA, USA). The Markov model is presented in Figure 1. Open surgery was compared to conventional laparoscopy and robotic surgery separately. The model was cycled per annum. Using GetData™ Software (Ver. O.11.0.), we digitized the disease-specific survival and the loco-regional recurrence curves in the study by Ramirez et al. [10], and used the Nelder–Mead Algorithm to fit exponential and Weibull distributions to the data to estimate the mortality and the progression rates, respectively. Surgery costs of open, laparoscopic, and robotic radical hysterectomy were taken from the National Inpatient Sample of the Healthcare Cost and Utilization Project, as published by Uppal et al. [23], and hence, data are relevant to USA pricing. All costs were discounted at 3%. The value of health benefits for each strategy (open radical hysterectomy versus minimally invasive surgery), from the payer’s perspective, was calculated using quality-adjusted life years (QALYs), which reflect both the quality and quantity of life lived. QALYs were calculated by multiplying the utility value associated with a given state of health by the number of years lived in that state, where a QALY of one reflects one year lived in perfect health and a QALY of zero represents death state. Incremental cost-effectiveness ratio was used for cost-effectiveness analysis. The incremental cost-effectiveness ratio is fundamentally the difference in costs of the different surgical approaches to radical hysterectomy, divided by the difference in outcomes, measured in QALYs, and represents the ratio between extra costs per extra unit of health benefit from the surgical treatment of cervical cancer. The incremental cost-effectiveness ratio was calculated by using the following formula: (average cost minimal invasive surgery—average cost open surgery)/(average QALY minimal invasive surgery—average QALY open surgery). One-way sensitivity analysis was conducted for all variables, in order to evaluate model uncertainties. Model probabilities and assumptions are presented in Table 1.

## 3. Results

Open radical hysterectomy compared to minimally invasive radical hysterectomy is cost-saving with an incremental cost-effectiveness ratio of USD −66 and USD −373 for laparoscopic radical hysterectomy and robotic radical hysterectomy, respectively, per QALY gained (Table 2). One-way sensitivity analysis was conducted for all variables. The tornado diagram (Figure 2) presents the influence of each parameter on the calculated incremental cost-effectiveness ratio. The most influential parameters that affect the incremental cost-effectiveness ratio are surgery costs, followed by the disutility involved with open surgery, and finally, disease-specific survival for each strategy.

## 4. Discussion

While introducing contemporary surgical techniques in surgical oncology, great caution must be taken in order to ensure their oncological safety but also their financial feasibility. The aim of the present study was to investigate the cost-effectiveness of open radical hysterectomy, compared to minimally invasive radical hysterectomy for early-stage cervical cancer, in light of the recently published data regarding the differential survival of patients undergoing open surgery compared to minimally invasive surgery. According to our results, open radical hysterectomy is more cost-effective compared to minimally invasive surgery and is, in fact, according to our data, cost-saving. This is true when comparing open surgery to both laparoscopic radical hysterectomy (ICER of USD −66/QALY gained) and robotic radical hysterectomy (USD −373/QALY gained) (Table 2). This means that patients with early-stage cervical cancer that have open radical hysterectomy not only fare better oncologically but also pose a lower financial burden, at current US pricing, on the health system, which needs to cover considerable treatment-related costs. The clinical significance of a negative ICER (USD −66 and −373) indicates that open radical hysterectomy not only achieves favorable health outcomes but also does so at a lower overall cost compared to laparoscopic surgery, thus offering a potentially cost-saving and effective option for improving patient care.

Historic, mainly retrospective and case–control data comparing the safety of minimally invasive radical hysterectomy to open radical hysterectomy did not demonstrate any survival difference between the two approaches. This, along with the many advantages of minimally invasive surgery, made minimally invasive radical hysterectomy an attractive alternative [6,7,28,29,30,31,32,33,34]. Even though no prospective, adequately powered trials examined the oncological safety of minimally invasive surgery for early-stage cervical cancer, this approach gained favor [11]. Shortly after the publication of the LACC trial, as well as other epidemiological studies and meta-analyses [10,11,17] that showed decreased overall survival for the minimally invasive surgical approach to cervical cancer treatment, the number of minimally invasive radical hysterectomies rapidly decreased [35]. Yet, shorter hospital stay, decreased need for blood transfusions, fewer surgical complications, as well as other cost-saving advantages of minimally invasive surgery do seem to outweigh the higher surgery-related costs involved with laparoscopic and especially robotic surgery [1,20,23]. These costs, combined with the updated survival data of cervical cancer patients, make the open approach more cost-effective compared to MIS radical hysterectomy.

One of the most influential parameters that affected the incremental cost-effectiveness ratio was the disutility value that accompanies open surgery (Figure 2). Hence, from only a cost-effectiveness analysis perspective, patients that would benefit from minimally invasive surgery for early-stage cervical cancer are only those who would find open surgery to cause an extreme negative effect on their quality of life (disutility). In a recent article published by Frumovitz et al., a secondary outcome analysis of the LACC trial, comparing quality of life among patients treated for open versus MIS for early-stage cervical, no differences in quality of life parameters were observed at baseline, 6 weeks after surgery, nor at 3 months after surgery [36]. From the later study, we can conclude that most probably, the disutility of open surgery is low, further supporting the results of our model.

The other most important parameters that affected the incremental cost-effectiveness ratio in our model were surgery costs (Figure 2). The costs of open surgery are mainly influenced by the higher surgery-related complications and longer hospital stay, and are likely to change only by meticulous surgical techniques and reduced complication rates. On the other hand, the costs of minimally invasive surgery, especially robotic surgery, are expected to decrease in the future, with the introduction of other robotic surgical platforms that would be available for clinical use. Lower procedure-related costs may tip the balance and make minimally invasive surgery more cost-effective.

As opposed to cervical cancer, minimally invasive surgery does not seem to jeopardize overall and disease-free survival among endometrial cancer patients [37]. Several theories were proposed to explain the findings of the LACC trial; these may be related to the use of uterine manipulators and or the colpotomy technique that may increase the risk of tumor spread during minimally invasive surgery. This may not be the case for endometrial cancer, which is usually confined to the endometrial cavity, and hence is less likely to spread during colpotomy. In any case, once the pathophysiological mechanism is understood which causes MIS to be oncologicaly inferior, compared to open surgery, for early-stage cervical cancer, and once survival rates are comparable, MIS may become the more cost-effective approach.

Our work has several drawbacks. The costs of minimally invasive surgery, including laparoscopic and robotic surgery, vary across reports. Nonetheless, in all reports, minimally invasive radical hysterectomy is more costly than open surgery [1,20,23]. We chose to use pricing reported by Uppal et al., who utilized data from the National Inpatient Sample of the Healthcare Cost and Utilization Project, which is the largest publicly available all-payer inpatient database in the United States, and hence seems the most comprehensive. The differences seen in survival between patients treated by minimally invasive and open surgery may not apply to patients with small, microscopic, or macroscopic tumors < 2 cm (2018 FIGO stage 1A1-1B1); for these subgroups of patients, the cost-effectiveness analysis may be different. The number of patients in this subgroup may be significant (in the LACC trial, 55% of patients with information on tumor size had tumors < 2 cm) [10].

Future research should aim at clarifying the causes of the survival differences observed in the LACC trial and point out specific subgroups of patients that would benefit from the obvious advantages that minimally invasive surgery has to offer. Examples of such patients include patients with small <2 cm tumors, or patients that have conization prior to definitive surgery. Survival data of these patients, integrated into our model, could show different results regarding cost-effectiveness.

## 5. Conclusions

Until further data are generated regarding the survival of patients with early-stage cervical cancer treated by minimally invasive surgery, at current pricing, open radical hysterectomy is cost-saving compared minimally invasive radical hysterectomy, both laparoscopic and robotic, and should be the preferred surgical approach.

## Figures and Tables

**Figure 1 cancers-15-04325-f001:**
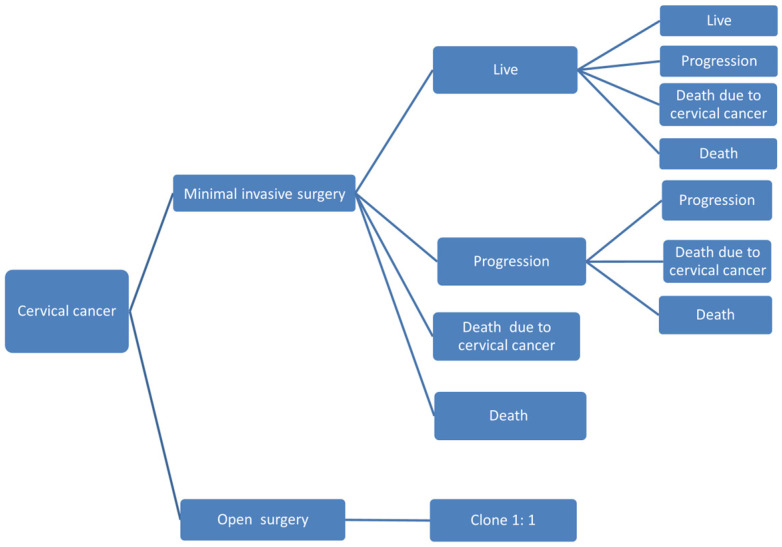
Markovian process decision analysis model: minimally invasive versus open surgery for early-stage cervical cancer.

**Figure 2 cancers-15-04325-f002:**
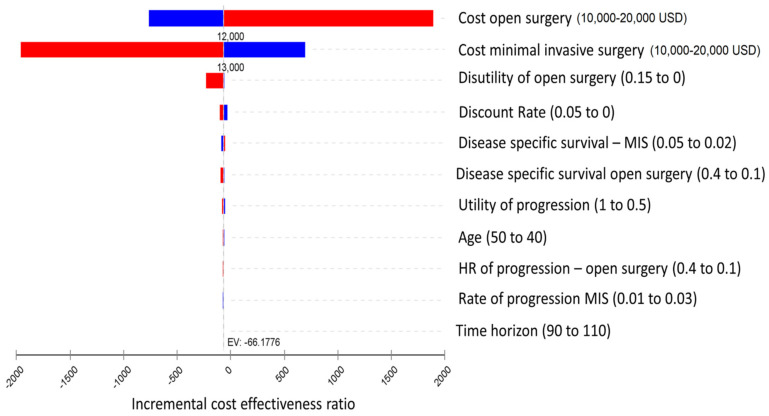
Tornado diagram—incremental cost-effectiveness ratio, minimally invasive surgery versus open surgery.

**Table 1 cancers-15-04325-t001:** Assumptions used in the model.

Description	Base	Low	High	Ref.
Age	45	40	50	assumption
disutility of Open surgery	0.02	0	0.4	[27]
HR of Disease-Specific Survival—Open Surgery	0.152	0.1	0.4	[10]
HR of Progression—Open Surgery	0.235	0.1	0.4	[10]
Discount Rate	0.03	0	0.05	assumption
Rate of Disease-Specific Survival—Minimally Invasive	0.015	0.01	0.02	[10]
Rate of Progression—Minimally Invasive	0.0216	0.01	0.03	[10]
Time Horizon—Until age	100	90	110	assumption
Utility of Progression	0.8	0.5	1	[27]
Cost of Laparotomy (Open Surgery)	12,624	10,000	20,000	[23]
Cost of laparoscopic surgery	12,873	10,000	20,000	[23]
Cost of Robotic surgery	14,029	10,000	20,000	[23]

**Table 2 cancers-15-04325-t002:** Cost-effectiveness analysis of minimally invasive (laparoscopic and robotic) radical hysterectomy compared to open hysterectomy for the treatment of early-stage cervical cancer.

**Strategy**	**Cost**	**Incremental Cost**	**Effect**	**Incremental Effect**	**Incremental Cost-Effectiveness Ratio**
Open radical hysterectomy	12,624		20.88		
Laparoscopic radical hysterectomy	12,873	249	17.12	−3.76	−66.18
**Strategy**	**Cost**	**Incremental Cost**	**Effect**	**Incremental Effect**	**Incremental Cost-Effectiveness Ratio**
Open radical hysterectomy	12,624		20.88		
Robotic radical hysterectomy	14,029	1405	17.12	−3.76	−373.41

## Data Availability

Not applicable.

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
