# Peer review of "Cost–Utility Analysis of Open Radical Hysterectomy Compared to Minimally Invasive Radical Hysterectomy for Early-Stage Cervical Cancer"

_cancers, 2023, doi:10.3390/cancers15174325_

Round 1

Reviewer 1 Report

Important note to the authors: On line 40, the paragraph started by the advantages of laparoscopic surgery over open surgery, but nothing mentioned about the advantages of open surgery versus the laparoscopic one. I think this is an important issue, reflecting impartiality and honesty of the scientific work.

Author Response

Thank you for this remark.

Some advantages of open surgery over laparoscopic surgery were added to the text (line 40-42). The main advantage of open radical hysterectomy for the treatment of cervical cancer is the improved survival of patients treated by the open technique. This is mention throughout the text and is the essence of this work (for exmple line 51-53).

Another advantage of open surgery over laparoscopic surgery is surgery costs. this is mentioned in the text (line 60-63).

Reviewer 2 Report

Thank you for this interesting paper evaluating costs effective data. I have several comments:

the data used for costing is all US based this should be noted early on in the paper as it is a major limitation, given that your sensitivity analysis shows this is the major factor. Are there any other costings from other countries? Can you comment further about an ICR of -$66 and is clinically significant?

You note that the issue of smaller cancers (<2 cm) may not have the same issues with laparoscopic surgery, do the data sets used note what percentage had these early cancers?

In the conclusions you note that the study shows open  surgery is oncologically safer however this it is just quoting other data, specifically the LACC trial and not a metaanlysis of multiple trials.

Author Response

Reviewer 2

Thank you for your important comments. Indeed, costing in this manuscript was done on US data. This limitation was added to the introduction (line 73), methods (line 89) and discussion (line 137). The Markovian model used is fully mentioned and adaptation to local costs can be easily made for reproducibility in other countries. Referring to pricing in other countries is beyond the scope of this work as all specific countries would have different and specific pricing.

The clinical significance of a negative ICER of -66 was better explained in the text (line 138-142).

The percent of patients with tumors smaller than 2 cm is not negligible, a comment was added to the discussion (line 195-196)

The conclusions were fixed and the sentence regarding the oncological safety of open surgery was removed from the conclusions (line 131-133)